# DNA Methylation at *ATP11A* cg11702988 Is a Biomarker of Lung Disease Severity in Cystic Fibrosis: A Longitudinal Study

**DOI:** 10.3390/genes12030441

**Published:** 2021-03-19

**Authors:** Fanny Pineau, Davide Caimmi, Sylvie Taviaux, Maurane Reveil, Laura Brosseau, Isabelle Rivals, Margot Drevait, Isabelle Vachier, Mireille Claustres, Raphaël Chiron, Albertina De Sario

**Affiliations:** 1LGMR, EA7402 University of Montpellier, 34093 Montpellier, France; fanny.pineau@live.fr (F.P.); sylvie.taviaux@orange.fr (S.T.); mauranereveil@gmail.com (M.R.); laura_brosseau@hotmail.fr (L.B.); mireille.claustres@inserm.fr (M.C.); 2CRCM, CHU Montpellier, 34090 Montpellier, France; dp-caimmi@chu-montpellier.fr (D.C.); m-drevait@chu-montpellier.fr (M.D.); r-chiron@chu-montpellier.fr (R.C.); 3IDESP, UMR INSERM, University of Montpellier, 34093 Montpellier, France; 4Equipe de Statistique Appliquée, ESPCI Paris, PSL Research University, UMRS1158, 75231 Paris, France; isabelle.rivals@espci.fr; 5CHRU de Montpellier, 34090 Montpellier, France; isabelle.vachier@medbiomed.fr; 6PhyMedExp, University of Montpellier, INSERM, CNRS, 34093 Montpellier, France

**Keywords:** cystic fibrosis, DNA methylation, sputum, longitudinal study, lung function, biomarker

## Abstract

Cystic fibrosis (CF) is a chronic genetic disease that mainly affects the respiratory and gastrointestinal systems. No curative treatments are available, but the follow-up in specialized centers has greatly improved the patient life expectancy. Robust biomarkers are required to monitor the disease, guide treatments, stratify patients, and provide outcome measures in clinical trials. In the present study, we outline a strategy to select putative DNA methylation biomarkers of lung disease severity in cystic fibrosis patients. In the discovery step, we selected seven potential biomarkers using a genome-wide DNA methylation dataset that we generated in nasal epithelial samples from the MethylCF cohort. In the replication step, we assessed the same biomarkers using sputum cell samples from the MethylBiomark cohort. Of interest, DNA methylation at the cg11702988 site (*ATP11A* gene) positively correlated with lung function and BMI, and negatively correlated with lung disease severity, *P. aeruginosa* chronic infection, and the number of exacerbations. These results were replicated in prospective sputum samples collected at four time points within an 18-month period and longitudinally. To conclude, (i) we identified a DNA methylation biomarker that correlates with CF severity, (ii) we provided a method to easily assess this biomarker, and (iii) we carried out the first longitudinal analysis of DNA methylation in CF patients. This new epigenetic biomarker could be used to stratify CF patients in clinical trials.

## 1. Introduction

Mutations in the *Cystic Fibrosis Transmembrane conductance Regulator* (*CFTR*) gene result in Cystic Fibrosis (CF, OMIM 219700), the most common life-shortening genetic disease in Caucasians (for a review see in [1]). The *CFTR* gene encodes a cAMP-dependent anion channel that is vital for bicarbonate (HCO3-) and chloride (Cl-) transport across the apical membrane of various epithelial cells. CFTR dysfunction or loss causes a severe imbalance in ion and water transport in multiple organs of the respiratory, digestive, and reproductive systems. The mutant CFTR protein is also responsible for an altered innate and adaptive immune function and for a defective dampening of the inflammatory response.

Pulmonary disease, which is characterized by chronic airway inflammation, recurrent infections, and progressive lung function decline, is the most common cause of morbidity and mortality in CF [2]. Gastrointestinal manifestations and exocrine pancreatic insufficiency affect body growth during development and challenge the nutritional status in adults. With age, a number of comorbidities become more prevalent among CF patients, namely, diabetes and liver disease [2].

No curative treatment is available so far, but the follow-up in specialized centers and symptom-targeted treatments (i.e., antibiotics, mucolytics, anti-inflammatory drugs, etc.) have greatly increased the life expectancy of CF patients over the last decades [2]. More recently, CFTR modulators have changed the therapeutic approach to CF, providing a substantial clinical benefit to patients carrying the responsive mutations. CFTR modulators include corrector molecules that stabilize protein maturation, and potentiator molecules that improve the transport of chloride by the CFTR protein [3]. Four approved treatments are now available, and more potential CFTR modulators are currently being developed. Overall, robust biomarkers are required to monitor the disease, guide treatments, stratify patients, and provide outcome measures in clinical trials.

Measurements of lung function, such as Forced Expiratory Volume in the first second (FEV_1_) and Forced Vital Capacity (FVC), are widely used for the follow-up of CF patients because of their correlation with the general state of the disease [4]. Changes in FEV_1_ are also the primary endpoints used to assess the clinical efficacy of new treatments in CF clinical trials [4,5]. 

In addition to clinical features, various biochemical parameters (i.e., interleukin-8, C-reactive protein, and neutrophil elastase levels) have been proposed as potential biomarkers for CF [6,7,8]. 

More recently, the evolving field of omics has generated molecular biomarkers for CF. Panels of genes or proteins were used to predict exacerbations [9,10], whereas circulating miRNAs predicted liver disease [11].

DNA methylation is a very stable epigenetic modification that can be easily measured in a small amount of genomic DNA, using suitable techniques for routine clinical testing [12]. DNA methylation profiling is a recent area of investigation in cystic fibrosis and few genome-wide datasets have been generated so far [13,14]. In a previous study, we showed that DNA methylation levels correlated with FEV_1_%, in nasal epithelial cell samples from CF patients [13]. Specifically, DNA methylation levels at 187 CpG dinucleotides were differentially methylated between CF patients with mild versus severe lung disease and explained 70% of FEV_1_% predicted variations [13]. These findings led us to hypothesize that DNA methylation levels could be used as prognostic biomarkers of pulmonary disease. In the present study, we outline a strategy to select putative DNA methylation biomarkers of lung disease severity in CF patients.

## 2. Materials and Methods

### 2.1. Cohorts and Samples Collection

The MethylCF cohort was previously described [13,15,16]. Briefly, it consists of 51 adult CF patients homozygous for the deletion of the phenylalanine 508 (p.Phe508del) variant in CFTR and 24 adult healthy controls with no history of airway disease. Nasal epithelial cell samples were collected from the inferior turbinate and genomic DNA was extracted [15,17].

The MethylBiomark cohort was built for the present study. It consists of 50 CF patients aged 12 to 30 years and carrying two severe (class I, II) *CFTR* variants in trans. Exclusion criteria included lung transplantation. Participants were enrolled in the CF center of Montpellier, France. Patients provided 1 to 4 sputum samples during a follow-up of approximately 18 months (Visits 1 to 4). Spontaneous sputum samples were collected by an experienced physiotherapist and were processed within 4 h after collection or stored at −80 °C for delayed treatment.

The FEV_1_% predicted, adjusted for age, was used to stratify CF patients into three groups (mild, intermediate, and severe) depending on the severity of their lung disease. Patients with FEV_1_% predicted values corresponding to the top and bottom quartiles were classified as mild and severe, respectively [18,19]. A pulmonary exacerbation was defined as an increase in respiratory and systemic symptoms requiring an oral or intravenous antibiotic treatment, as validated by experienced physicians participating in the study.

The study was approved by the local institutional review board (CPP Sud Méditerranée III, #2016.09.02bis) and registered at clinicaltrial.gov under reference #NCT02976714. Informed written consent was obtained from all participants. In case of pediatric patients, the minor and both parents signed the written consent.

### 2.2. Sputum Samples Processing

Mucus plugs were separated from saliva with forceps in a Petri dish and were treated with 6–13 mL of 1/10 Sputolysin^®^ (Calbiochem, La Jolla, CA, USA), depending on their size. The solution was homogenized for 15 min at 37 °C with gentle shaking. PBS was then added in the same volume as Sputolysin^®^. After centrifugation, the pellet was resuspended in 1 mL PBS. The total cell number, the proportion of contaminating buccal squamous cells and the number of necrotic cells were assessed using a hemocytometer. Differential cell count was performed after Kwik-Diff staining (kit #9990700, Thermo Fisher Scientific MA USA) on cytospins. Genomic DNA was extracted from around 3 × 10^6^ cells using the QIAamp DNA Mini kit (#51304, Qiagen, MD, USA). We followed the manufacturer’s instructions from the “Blood or Body Fluid Spin Protocol”, except that samples were incubated for 90 min in buffer AL at 56 °C with agitation at 300 rpm in a Thermomixer Comfort (Eppendorf Hamburg, Germany). DNA was eluted in 100 µL Tris-EDTA buffer and stored at −20 °C. The flow chart and the detailed protocol are available (Appendix A
Appendix A).

### 2.3. Pyrosequencing DNA Methylation Analysis

Sputum genomic DNA was treated with sodium bisulfite using the EpiTect kit (#59104, Qiagen) according to the manufacturer’s protocol. Pyrosequencing was carried out as previously described [15]. Briefly, PCR products were amplified with the PyroMark PCR kit (#978703, Qiagen) and purified with Streptavidin Sepharose HP™ (#17-5113-01, GE Healthcare) using a PyroMark Q24 Workstation. Pyrosequencing reactions were done with Gold Q24 reagents (#970802, Qiagen) using a PyroMark Q24 (Qiagen). We tested the signal linearity of pyrosequencing assays using mixtures of methylated and unmethylated genomic DNA (0, 20, 40, 60, 80, and 100%; from EpiTect Control DNA #59655 and #59665, Qiagen) and calculated standard errors from three replicates. Pyrosequencing assays were approved when the affine relation between expected and observed DNA methylation displayed a slope > 0.75 and a R2 > 0.96. DNA methylation measurements were done at least in duplicate. PCR and sequencing primers are provided in Appendix A.

### 2.4. Statistical Analyses

Differential DNA methylation analysis was carried out with Wilcoxon tests. Correlations between DNA methylation and clinical variables at a given time point (Visits 1 to 4) were calculated using Spearman’s correlation coefficient adjusted for age and sex. To assess the strength of the association along all 4 visits, linear mixed models of the form y ~ x + age + sex + 1|patient where estimated, where y denotes the methylation of CpG methylation, x the clinical variable of interest, and 1|patient a random intercept modeling the patient effect. The strength of the association was assessed through the *p*-value of the fixed effect of the clinical variable x. These models being parametric, the normality of the residuals was checked using the Shapiro–Wilk test.

### 2.5. Enhancers Prediction

To assess the chromatin organization in the regions of interest, we retrieved H3K27 acetylation (GSM906395), H3K4 mono-methylation (GSM910572), and H3K4 tri-methylation (GSM915336) ChIP-seq data for a lung sample from the Roadmap Epigenomics Project (http://www.roadmapepigenomics.org) (accessed on 7 March 2021). Reads were aligned to the reference human genome build hg19 using Bowtie2 [20]. For peak calling we used MACS2 [21] and default parameters with the exception of no-lambda option because no input was available. Moreover, to detect H3K4 mono-methylation peaks, we used the broad option. Finally, by assessing the histone profile in the regions of interest, we annotated the active enhancers. A CpG site associated with H3K27 acetylation and H3K4 mono-methylation, but not with H3K4 tri-methylation, was considered associated with an active enhancer [22].

## 3. Results

### 3.1. MethylBiomark Cohort and Biobank

The 50 CF patients of the MethylBiomark cohort were enrolled in the CF center of Montpellier (France) and participated for approximately 18 months, in the framework of their regular follow-up. We included 36 adult (>18 years) CF patients, excluding individuals aged more than 30 years to prevent an overrepresentation of mild phenotypes. We also included 14 teenagers aged 12 to 18 years. In terms of genotypes, the MethylBiomark cohort is representative of the CF population in the south of France, with 72% of homozygous p.Phe508del patients and 26% of patients with heterozygous p.Phe508del genotype. Considering their FEV_1_% predicted value at the baseline visit (V1), 11 patients were classified as having mild lung disease (top quartile), 29 had intermediate lung disease, and 10 had severe lung disease (bottom quartile). All patients presented exocrine pancreatic insufficiency and 12% had diabetes. Finally, 42% of CF patients were treated with the modulator lumacaftor/ivacaftor (lumacaftor is a corrector molecule that stabilizes the CFTR protein maturation, and ivacaftor is a potentiator of the channel function) [23].

Sputum samples were collected approximately every 6 months, and clinical data (i.e., spirometric measurements, presence of germs, exacerbations, and treatments) were recorded on the same day. To treat sputum samples and extract genomic DNA, we adapted and combined previously well-established methods to provide a streamlined strategy that could be routinely used for patient follow-up. Briefly, to minimize contamination with squamous cells from buccal mucosa, we selected mucus plugs from saliva and dispersed them with Sputolysin^®^. About 75,000 cells were used for quality controls. The remaining cells provided aliquots for DNA extraction. We biobanked samples containing (i) a minimal total amount of 200,000 cells, (ii) less than 15% of buccal squamous cells, and (iii) no more than 20% of necrotic cells. The MethylBiomark biobank consists of 144 validated sputum and matched genomic DNA samples: 43 from baseline visit (V1), 40 from visit 2, 32 from visit 3, and 29 from visit 4. The average cell composition of the sputum samples was 74% of neutrophils, 15% of macrophages, 5% of buccal squamous cells, and 6% of other cell types (eosinophils, red blood cells, epithelial cells, and lymphocytes) (Figure 1). Genomic DNA was extracted from the sputum samples with a median yield of 19.1 µg [Q1 = 9.8; Q3 = 31.9]. Four samples contained no mucus plugs and twelve samples were withdrawn because of contamination with buccal squamous, necrotic, or circulating blood cells. Finally, eight sputum samples failed to provide a sufficient amount of DNA (<2 μg). Overall, we built a CF cohort and genomic DNA biobank to be used in the present pilot study, as well as in future studies.

### 3.2. Biomarker Discovery in Nasal Epithelial Cell Samples from the MethylCF Cohort

During a previous study, we generated a genome-wide DNA methylation dataset in nasal epithelial cell samples from CF patients and healthy controls (MethylCF cohort) [13]. Herein, we used this methylation dataset to select potential biomarkers of lung disease severity in cystic fibrosis. We combined differential DNA methylation and correlation analyses and selected top CpG sites. CpG dinucleotides were considered differentially methylated when the difference between the median β value for mild patients and the median β value for severe patients (∆β) was >0.1, with an associated *p*-value < 0.01. The β value ranges from 0 (no methylation) to 1 (full methylation). Correlations between DNA methylation and FEV_1_% predicted were calculated through Spearman test adjusted for age and sex. To avoid selecting CpG with DNA variations that could be confounded with background noise, we used a threshold on the methylation median absolute deviation (MAD). CpG sites were considered significantly correlated with FEV_1_% predicted when the Spearman adjusted *p*-value was <0.05 and the methylation MAD was >0.01. We selected seven potential biomarkers of lung disease severity in cystic fibrosis: all CpG were differentially methylated between mild and severe CF patients and four of them also correlated with FEV_1_% (Table 1). Six CpG sites were associated with a protein-coding gene and one CpG site was intergenic.

### 3.3. Biomarker Assessment in Sputum Samples from the MethylBiomark Cohort

Seven CpG markers were re-assessed in sputum samples from the prospective longitudinal MethylBiomark cohort. DNA methylation was measured by pyrosequencing in 43 sputum samples collected at the baseline visit (V1). Next, we correlated DNA methylation of each CpG site with continuous or binary phenotypic traits of the cohort: lung disease severity, lung function metrics, and presence of chronic infections and treatments (Figure 2 and Appendix A). The clinical data were filed on the same day the sputum samples were collected (Table 2).

Of interest, DNA methylation of *ATP11A* (cg11702988) positively correlated with lung function (FEV_1_ and FVC) and BMI, and negatively correlated with lung disease severity (mild, intermediate, and severe CF patients) and with the presence of chronic *P. aeruginosa*. In addition, this methylation marker negatively correlated with the number of exacerbations that the patient had in the 24-month time after the baseline visit, but did not correlate with the presence of an exacerbation when the sample was collected.

No correlation was found between the six other CpG sites and lung function metrics, or with disease severity.

### 3.4. Longitudinal Analysis of DNA Methylation at ATP11A (cg11702988) in Sputum Samples

To provide a longitudinal survey in CF patients, we assessed DNA methylation at *ATP11A* (cg11702988) in sputum samples collected at four visits (V1 to V4) during an 18-month follow-up. Each visit was approximately six months apart from the previous one. Overall, we analyzed 144 sputum samples from 50 CF patients of the MethylBiomark cohort (Table 2).

In nasal epithelial cell samples from the MethylCF cohort, *ATP11A* (cg11702988) was less methylated in severe CF patients than in mild or intermediate patients (Table 1). We observed the same trend in sputum samples collected from the MethylBiomark cohort (Figure 3a).

Next, we correlated DNA methylation of the *ATP11A* (cg11702988) marker and clinical data at each time point and we found a positive correlation with lung function at four successive time points (Figure 3b and Appendix A). Of interest, we also found a positive correlation between DNA methylation and BMI and a negative correlation with *P. aeruginosa* chronic infection.

Then, the correlation analyses at each time point were supplemented by a longitudinal analysis taking all four visits into account using linear mixed models. We assessed the strength of the association between methylation at *ATP11A* and each clinical variable through the significance of fixed effect of the latter one (Appendix A). Again, *ATP11A* (cg11702988) positively correlated with lung function and negatively correlated with lung function severity, the number of exacerbation, and the presence of *P. aeruginosa* chronic infection. The effects of age and sex never reached a statistical significance (with a type one error risk of 5%).

To conclude, the correlation between DNA methylation at *ATP11A* (cg11702988) and variables that are considered good indicators of the clinical status of CF patients was replicated at four time points and longitudinally over approximately 18 months.

### 3.5. The cg11702988 (ATP11A) Maps to a Predicted Enhancer in the Lung

The cg11702988 (chr13:113,422,177 Hg19) maps to intron 1 of *ATP11A*. This gene is ubiquitously expressed in human tissues, with a maximum in alveolar cells type 1 and 2 in the lung, according to transcriptomic datasets in human tissues and single cells (the Genotype-Tissue Expression (GTEx) Project and Human Protein Atlas, respectively) (Figure 4a) [24,25]. To assess the chromatin structure around this CpG site, we retrieved ChIP-seq datasets from the Roadmap Epigenomics Project and analyzed three histone modifications that are predictive of active *cis*-regulatory sequences [22]. In adult lungs, the cg11702988 was associated with high levels of H3K27 acetylation and H3K4 mono-methylation, in the absence of H3K4 tri-methylation (Figure 4b). To conclude, the cg11702988 was associated with histone modifications suggestive of an open chromatin and potential enhancer activity in the adult lung.

## 4. Discussion

Robust biomarkers are required for CF monitoring, to guide treatments, stratify patients, and as outcome measures in clinical trials. In the present study, we outline a strategy to identify DNA methylation markers in sputum samples from CF patients. DNA methylation is a very robust epigenetic mark and can be easily and quantitatively assessed from few hundred nanograms of DNA. Thus, DNA methylation biomarkers have great potential in terms of disease diagnosis or prognosis and prediction of drug response [12].

In this pilot study, we preselected seven potential biomarkers in the MethylCF cohort using an array hybridization DNA methylation dataset and successfully replicated a biomarker in the MethylBiomark cohort. To the best of our knowledge, we carried out the first longitudinal analysis of DNA methylation in CF patients. DNA methylation level at the cg11702988 dinucleotide was lower in CF patients presenting a severe lung disease, positively correlated with lung function and BMI, and negatively correlated with the presence of a *P. aeruginosa* chronic infection and with the number of exacerbations. The correlation with these clinical features was stable over time, including during pulmonary disease exacerbations. These findings lead us to suggest that cg11702988 methylation reflects the disease severity, rather than its activity. It could be used to stratify patients increasing the power of small clinical trials. We analyzed cg11702988 DNA methylation in adult CF patients and 12- to 18-year-old children. In future studies, it will be interesting to assess whether the correlation between this epigenetic biomarker and disease severity is substantiated in children under age 12 years. If this is the case, the new biomarker could be useful to identify high-risk patients at an early stage, when spirometric measures are almost normal, but the inflammatory response is already altered.

In the discovery step, we selected potential methylation biomarkers in patients homozygous for the very common p.Phe508del mutation. This mutation affects the stability of the CFTR protein and its trafficking to the apical membrane of the epithelial cells, causing a severe phenotype [26]. In the validation step, to target a larger population of CF individuals, we enrolled patients carrying two severe *CFTR* mutations in trans. We successfully replicated the DNA methylation biomarker, showing that it is informative to CF patients with a larger spectrum of severe mutations. In future studies, the marker should be assessed in patients representative of all the most frequent CFTR mutations.

To provide a minimally invasive way to assess the biomarker, we validated the cg11702988 methylation in spontaneous sputum samples. Spontaneous and induced sputum samples are widely used to assess inflammation, detect pulmonary infections, and monitor drug response in cystic fibrosis [27]. It may be argued that treatment with CFTR modulators makes sputum collection difficult, because improvement of patient condition decreases mucus secretion. However, in the present study, we successfully collected sputum samples from lumacaftor/ivacaftor-treated patients. Moreover, induction can be implemented when spontaneous sputum is difficult to collect. Finally, we replicated the biomarkers using pyrosequencing (Pyromark, Qiagen), a cost- and time-effective technique that is already used in clinical setting [28]. To conclude, the present study has disclosed an epigenetic biomarker of CF severity in a wide range of patient ages and among CF patients carrying various class I and II *CFTR* mutations.

Biomarkers do not need to have biological relevance. However, because the cg11702988 dinucleotide maps to a predicted enhancer in the *ATP11A* gene, it may be questioned (i) whether methylation of this CpG has an impact on gene transcription and (ii) whether *ATP11A* is a CF modifier gene. *ATP11A* (ATPase Phospholipid Transporting 11A) encodes a flippase that maintains the asymmetry in the lipid bilayer of cell and organelle plasmic membranes by translocating phosphatidylserine and phosphatidylethanolamine from the outer to the inner leaflet [29]. The asymmetrical distribution of phospholipids is important for a variety of cellular functions including vesicle exocytosis, apoptosis, and cellular signaling. Specifically, Van der Mark et al. showed that the ATP11A protein is essential to decrease the inflammatory response through endotoxin-induced internalization of TLR4 [30]. Finally, a SNP (rs1278769) in the 3′UTR of *ATP11A* was associated with fibrotic idiopathic interstitial pneumonias in GWAS [31]. Overall, the function of this protein and results from previous genetic studies are consistent with ATP11A playing an important role in chronic lung diseases. However, functional studies are needed to address the specific role of ATP11A in the CF physiopathology. Additionally, ATP12A, a H+/K+-ATPase localized at the apical membrane of airway epithelial cells, was shown to be a pathogenic factor in cystic fibrosis [32].

Another important issue is whether low DNA methylation at cg11702988 is the cause or the consequence of severe disease in CF patients. In the general population, DNA methylation levels depend mainly upon non-inheritable environmental factors, in a smaller proportion on heritable genetic factors, and also on stochastic events [33]. Among non-inheritable factors responsible for DNA methylation levels, there are lifestyle factors and, in the case of CF patients, the disease itself. Permanent and excessive inflammation, chronic exposure to bacterial infections, and exocrine pancreatic insufficiency can shape DNA methylation [34]. Methylation at the cg11702988 was low in two types of tissues from severe CF patients (nasal epithelial cell and sputum samples) and was stable overtime in sputum samples. DNA methylation neither correlated with patient age, nor was altered by exacerbations. These findings lead us to suggest that DNA methylation at cg11702988 is genetically driven. Furthermore, although no methylation quantitative trait locus was associated with this dinucleotide in the meQTLdb and Pancan_meQTL databases [35,36], we still cannot rule out the possibility that an unknown *cis-* or *trans-* sequence variant affects DNA methylation at this CpG site.

This study has limitations. Even though we identified a new epigenetic biomarker of lung disease severity in CF, additional validation studies in larger cohorts are required to translate the biomarker to the clinical use. Herein, we selected markers in nasal epithelial cell samples and replicated them in sputum samples. Likely, more markers could be successfully identified, if discovery and replication were carried out in the same tissue type.

## 5. Conclusions

DNA methylation studies in cystic fibrosis are in their preliminary stages but show promise. Herein, (i) we identified a DNA methylation biomarker that correlates with cystic fibrosis severity, (ii) we provided a method to easily assess this epigenetic marker, and, to the best of our knowledge, (iii) we carried out the first longitudinal analysis of DNA methylation in CF patients. This new epigenetic marker could be used to stratify CF patients in clinical trials and also to predict high-risk patients.

## Figures and Tables

**Figure 1 genes-12-00441-f001:**
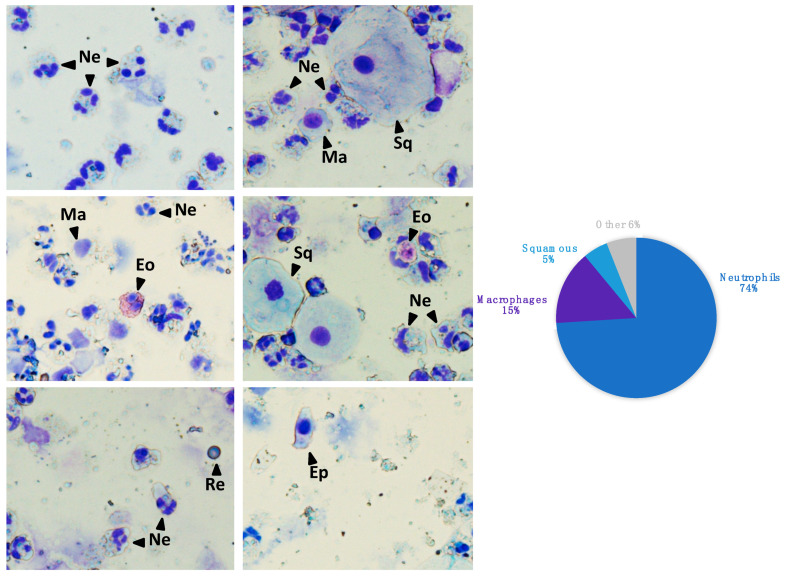
Cell count of sputum samples from the MethylBiomark cohort. Neutrophils (Ne) were dominant, followed by macrophages (Ma), small percentages of buccal squamous cells (Sq), and other cell types—eosinophils (Eo), red blood cells (Re), epithelial cells (Ep), and lymphocytes. Cytospins were stained with Kwik Diff.

**Figure 2 genes-12-00441-f002:**
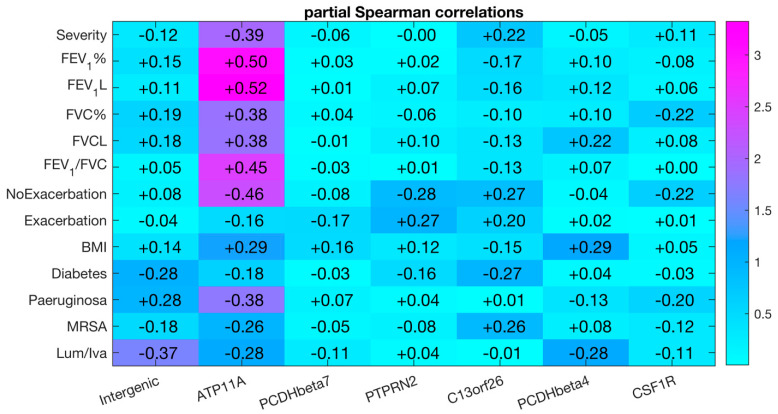
Age- and sex-adjusted Spearman correlations between DNA methylation at seven CpG sites and continuous or binary phenotypic traits of the MethylBiomark cohort. DNA methylation was measured by pyrosequencing in sputum samples collected at V1. Severity refers to the groups of mild, intermediate, or severe CF patients. NoExacerbation, number of exacerbations in 24 months after V1. Exacerbation, presence of exacerbation at V1. *MRSA*, methicillin-resistant *Staphylococcus aureus*. Lum/iva, lumacaftor/ivacaftor treatment. The color ladder represents minus the decimal logarithm of the *p*-value associated with the partial Spearman correlations.

**Figure 3 genes-12-00441-f003:**
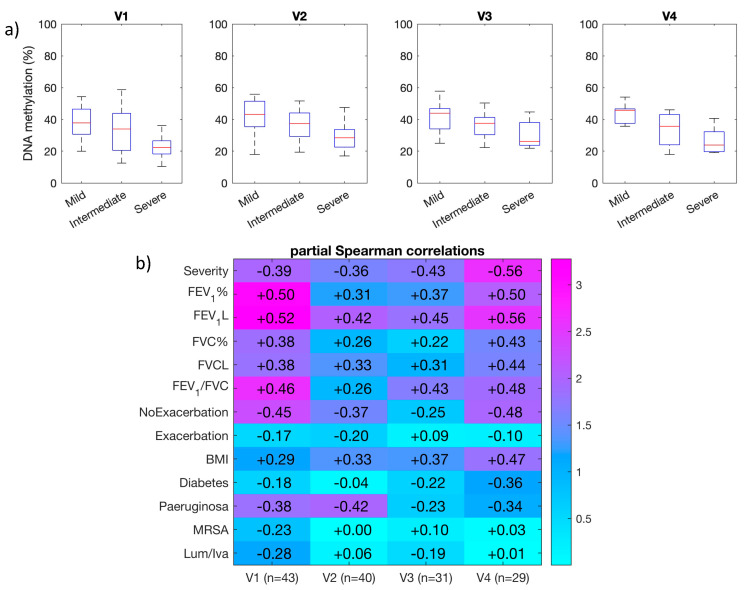
DNA methylation of *ATP11A* (cg11702988) in sputum samples from the MethylBiomark cohort at four time points (V1, V2, V3 and V4). (**a**) DNA methylation of mild, intermediate, and severe CF patients. (**b**) Age- and sex-adjusted Spearman correlations between DNA methylation and continuous or binary phenotypic traits. Severity refers to the groups of mild, intermediate or severe CF patients. NoExacerbation, number of exacerbations in 24 months after V1. Exacerbation, presence of exacerbation at the visit. *MRSA,* methicillin-resistant *Staphylococcus aureus*. Lum/iva, lumacaftor/ivacaftor treatment. The color ladder represents minus the decimal logarithm of the *p*-value associated with the partial Spearman correlations.

**Figure 4 genes-12-00441-f004:**
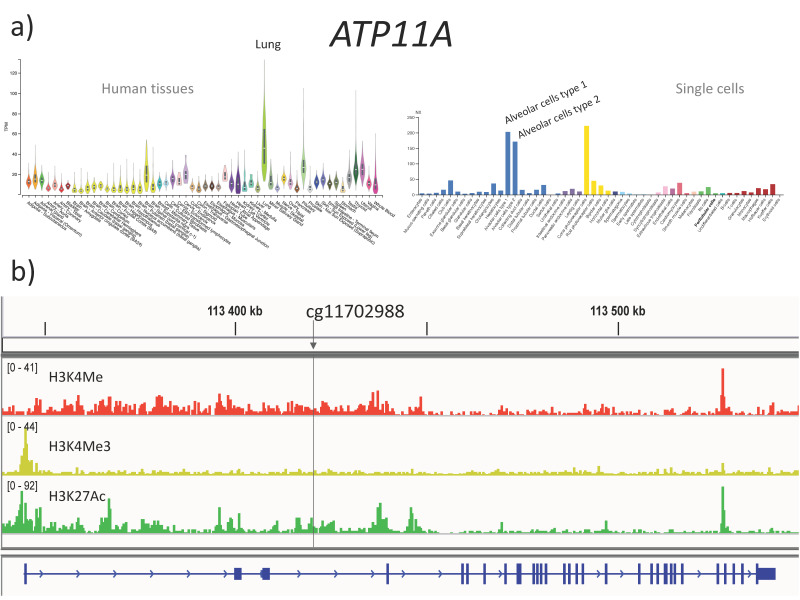
*ATP11A* gene. (**a**) Gene expression in human tissues (Genotype-Tissue Expression (GTEx) Project [24]) and in single cell RNA-seq datasets (Human Protein Atlas [25]). (**b**) Histone modifications in adult lung are represented with IGV. The vertical line shows the cg11702988.

**Table 1 genes-12-00441-t001:** Seven biomarkers were selected using a DNA methylation dataset generated in nasal epithelial cell samples from the MethylCF cohort.

		Correlation with FEV_1_%		Differential DNA Methylation	
CpG	Gene	Coefficient	*p*-Value	∆β (Mild-Severe) ^†^	*p*-Value ^§^
cg10582608	Intergenic	0.14	4.53 × 10^−1^	0.19 (0.34–0.15)	1.9 × 10^−3^
cg11702988	*ATP11A*	0.14	4.59 × 10^−1^	0.43 (0.78–0.35)	3.2 × 10^−3^
cg17735593	*PCDHβ7*	0.20	2.77 × 10^−1^	0.17 (0.66–0.49)	5.3 × 10^−3^
cg23299919	*PTPRN2*	0.45	1.23 × 10^−2^	0.25 (0.45–0.20)	5.6 × 10^−4^
cg08379987	*C13orf26*	0.67	5.70 × 10^−5^	0.19 (0.57–0.38)	3.2 × 10^−3^
cg06048354	*PCDHβ4*	0.39	3.16 × 10^−2^	0.16 (0.41–0.25)	6.7 × 10^−3^
cg05524038	*CSF1R*	0.49	5.83 × 10^−3^	0.13 (0.79–0.66)	1.2 × 10^−4^

Age and sex adjusted Spearman correlations between DNA methylation and FEV_1_%. ^†^ β (β) values from 450 K array, where 1 corresponds to full methylation and 0 to no methylation. ∆β = median β mild CF patients-median β severe CF patients. ^§^ Wilcoxon’s test.

**Table 2 genes-12-00441-t002:** MethylBiomark cohort: demographic and clinical data (number or percentage or median (iqr)) of patients who provided validated sputum samples at visit 1 to 4.

	V1	V2	V3	V4
Age, year	21.5 (7.8)	21.7 (9.4)	22.4 (8.6)	22.5 (10.2)
Sex ratio (M:F)	26:17	22:18	18:14	16:13
*CFTR* genotype				
p.Phe508del-p.Phe508del	34	30	25	19
p.Phe508del-other variant ^¶^	9	9	7	9
p.Arg553*-p.Trp1282*	0	1	0	1
Mild	8	9	8	6
Intermediate	26	22	16	15
Severe	9	9	8	8
Weight, kg	57.0 (15.0)	56.0 (15.5)	58.0 (17.5)	54.0 (11.8)
Height, cm	166.0 (15.0)	167.0 (14.5)	169 (14.5)	167 (12.5)
Body mass index, kg/m^2^	19.9 (3.0)	19.7 (2.9)	20.3 (3.4)	19.8 (2.1)
FEV_1_,% predicted	63.5 (31.5)	66.0 (36.7)	65.0 (41.5)	59.0 (40.8)
FEV_1_, liters	2.2 (1.0)	2.1 (1.5)	2.1 (1.5)	2.0 (1.5)
FVC,% predicted	81.0 (21.5)	79.0 (27.5)	83.0 (27.0)	82.0 (27.5)
FVC, liters	3.4 (1.0)	3.2 (1.2)	3.4 (1.6)	3.1 (1.6)
Exacerbation ^#^	12%	35%	35%	60%
Pancreatic insufficiency	100%	100%	100%	100%
Diabetes ^†^	12%	20%	19%	20%
Atopy	35%	40%	42%	30%
*Pseudomonas aeruginosa* ^‡^	72%	65%	61%	60%
*Aspergillus fumigatus* ^‡^	26%	25%	23%	20%
MRSA ^‡^	16%	22%	19%	20%
Other germs ^§^	81%	75%	77%	70%
Azithromycin	79%	72%	68%	80%
Tobramycine	30%	22%	16%	20%
Colistin	58%	55%	55%	60%
Aztreonam	19%	20%	19%	10%
Nasal corticosteroid	30%	37%	35%	40%
lumacaftor/ivacaftor	47%	42%	45%	30%

FEV_1_: Forced Expiratory Volume in one second, FVC: Forced Vital Capacity, MRSA: methicillin-resistant *Staphylococcus aureus*. ^¶^ p.Gln542* (5 patients), p.Arg553* (2 patients), p.Trp846*, p.Tyr122*, p.Gln685Thrfs*4, p.Asn1303Lys, and deletions in intron 2 (c.54-5811_164+2186delins182) or 17a (c.2989-908_3085delinsGACAG). ^#^ A pulmonary exacerbation was defined as an increase in respiratory and systemic symptoms requiring an oral or intravenous antibiotic treatment. ^†^ The presence of diabetes was determined on the basis of an abnormal oral glucose tolerance test. ^‡^ CF patients were classified as chronically infected by *P. aeruginosa*, MSRA, and/or *A. fumigatus* whenever they had three consecutive positive sputum cultures after antibiotic treatment. ^§^ Other germs include methicillin-sensitive *Staphylococcus aureus*, *Achromobacter xyloxidans*, *Mycobacterium abscessus*, *Stenotrophomonas maltophilia*, *Mycobacterium avium*, *Burkholderia cepacia*, *Inquilinus limosus*, and *Acinetobacter*.

## Data Availability

The genome-wide DNA methylation dataset from nasal epithelial cell samples are available in Gene Expression Omnibus (GEO) with accession number GSE101641.

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
