# Peer review of "DNA Methylation at ATP11A cg11702988 Is a Biomarker of Lung Disease Severity in Cystic Fibrosis: A Longitudinal Study"

_genes, 2021, doi:10.3390/genes12030441_

Round 1

Reviewer 1 Report

Pineau et al. present a strategy to discover DNA methylation biomarkers of cystic fibrosis lung disease. Differential methylation of cg11702988 and positive correlations with lung function and BMI are reported. Furthermore, negative correlations with disease severity, chronic infection, and number of exacerbations are discovered. The study, to the best of the author’s knowledge, is the first longitudinal analysis of DNA methylation in CF patients. The manuscript contains sound science, but has questionable clinical impact.

  • Can the authors comment further on the current state of DNA methylation biomarkers as tools in the clinic for cystic fibrosis? To the best of my knowledge, such biomarkers are not used. Is this due to such biomarkers not being well studied yet, or is it a reflection of clinicians potentially not finding such biomarkers informative?
  • In the discussion, the point is raised that it will be interesting to see if the correlation between biomarker and disease severity is substantiated in children under 12 years of age. If so, the biomarker could be used to identify high-risk patients at an early age. This appears to be the most powerful potential clinical use of such biomarkers. However, since the correlation has not been substantiated yet, is it likely that clinicians would find this biomarker useful? There are more defined and developed biomarkers to gauge disease severity, chronic infection, BMI, lung function, etc. in teens and adults. Is it fair to say that the clinical benefits of the discovered biomarker are heavily dependent on whether such correlations are also present in children?
  • The authors claim that the biomarker is informative to CF patients with a large spectrum of severe mutations. Is this accurate? Only four of the 10 most frequent CFTR mutations are represented in the study. Only Phe508del is well-represented. Table 2 states that G542X is present in only five patients, R553X in two, W846X in one, etc. All but one patient have at least one F508del mutation. Therefore, can you really conclude that the biomarker is informative in a large spectrum of severe mutations? Even when the common W1282X, G551D, R1162X, etc. mutations have not been studied in even a single patient?
  • Related to the previous question, Table 2 uses different nomenclature for Phe508del and the other mutations. The most recent accepted nomenclature for G542X, for example, is Gly542X. R553X is Arg553X, 2184insA is Gln685ThrfsX4, etc. Please use the current names and not the legacy names. See the Cystic Fibrosis Mutation Database on sickkids.on.ca for updates.
  • Functional studies are indeed needed to address the specific role of ATP11A in CF physiology. It might be worth mentioning the role of fellow ATPase ATP12A in CF physiology (Shah et al., Science (2016)). Although ATP11A and ATP12A are found in different classes of the ATPase family of enzymes, there is similarity in their mechanisms/roles. It is also worth mentioning that ATP11A was included in the large-scale siRNA screen for CFTR effectors performed by Tomati et al., Molecular Bases of Disease (2018) and knockdown was not found to affect plasma membrane targeting and function of Phe508del. Lastly, it would be interesting to view scRNA-seq data (at Human Protein Atlas or elsewhere) to determine which pulmonary cell types express ATP11A.
  • The authors state that they did not find methylation QTL associated with the dinucleotide and therefore the DNA methylation at cg11702988 might not be genetically driven. Therefore, is it likely that the disease itself (excessive inflammation, bacterial infections, etc.) is responsible for the methylation? If so, then it is possible that such methylation would not be observable in young patients who have not yet experienced permanent inflammation, chronic infections, etc., even further limiting the clinical use of the biomarker.

Overall, the discovery of the biomarker is scientifically interesting. However, its use as a clinical biomarker is questionable. A significant amount of research must be performed before the biomarker could be used to stratify patients in the clinic. It is still unclear which demographics of CF patients would benefit from such a clinical biomarker.

Minor comments:

  • When referring to the gene, “Cystic Fibrosis Transmembrane conductance Regulator” and “CFTR” should be italicized (g., see line 1 of introduction).
  • Clarify the following sentence: “The mutant CFTR protein is also responsible for an altered innate and adaptive immune function and for a defective down-regulation of the inflammatory response once this one has been activated.” It is unclear what is meant by “once this one has been activated.” What is “this one” referring to?
  • Reword the following: “…is the first cause of morbidity and mortality in CF.” Pulmonary disease is the most common cause of morbidity and mortality in CF, but is not always the cause and therefore is not always the “first.”
  • “Responding” should be changed to “corresponding” in the following sentence: “…providing a substantial clinical benefit to patients carrying the responding mutations.”
  • FEV1 and FEV1 (subscripted 1) are used throughout the manuscript. Either is acceptable, but use only one consistently throughout the paper. Likewise, FEV1 and FEV1% are used throughout the manuscript. As FEV1 is measured as a percentage, the % symbol is not needed and should be removed.
  • Change how numbers are presented in scientific notation from decimal point to “x”. For example, 3.106 to 3 x 106.
  • Change “1 h 30 min” to “90 min”.
  • Throughout the manuscript, sometimes there a space between a number and the percent symbol and in other instances no space is present. Remove the space between number and symbol throughout.
  • In section 3.2, the β value should be defined. β is described in the footnotes of Table 1, but should also be mentioned in the text.
  • As the manuscript is currently formatted, the top of Table 1 is cut off from the rest (between pages 5-6). Ensure that this is corrected in the final manuscript. The same goes for Table 2 and the “Discussion” header.
  • In section 3.5, the abbreviation “NEC” should be defined.
  • In section 3.6, it is stated that public databases were used to retrieve ChIP-seq datasets. Please identify these databases in the text.
  • Can you provide the numerical values for the ChIP-seq peaks seen at cg11702988? For example, the peak seen for H3K27Ac is obvious, whereas the peak at H3K4Me, while reported in the text as high, does not appear to be very noticeable. What is the threshold for “high” levels?
  • Section 3.6: “In adult lungs, the cg11702988 was associated with high levels of H3K27 acetylation and H3K4 mono-methylation, in the absence of H3K4 tri-methylation (Figure 4b). To conclude, the cg11702988 was associated with histone modifications suggestive of an open chromatin and potential enhancer activity in the adult lung.” Please provide a reference citing why H3K27 acetylation, H3K4 mono-methylation, and the absence of H3K4 tri-methylation are suggestive of enhancer activity.
  • In the legend of Figure 4, “GTex” should be changed to “GTEx”. The GTEx project should also be cited.
  • There have been other epigenome-wide DNA methylation studies in the CF lung that are worth mentioning and citing. For example, Chen et al., Clinical Epigenetics (2018).

Reviewer 2 Report

In the study by Pineau and coll., the status of methylation sites in cystic fibrosis patients is analyzed to find possible correlation with the clinical condition. Authors find a correlation between an intron site in ATP11A gene and various parameters including disease pulmonary function, pulmonary exacerbations, bacterial colonization. More precisely, methylation is increased in milder patients suggesting that suppression of ATP11A expression and/or function has a protective effect. This methylation site could therefore represent a useful biomarker. I have only a few comments for the authors.

Results section, end of first paragraph: authors quote reference 21 to support their description of potentiators and correctors mechanism of action. However, reference 21 is a clinical study. Authors should instead cite original articles describing how potentiators and corrects act on defective CFTR channel gating and trafficking.

Authors may quote and comment the article by Vincent et al. (Cell and Molecular Life Sciences 74:751-730, 2017) about phospholipid flippases, including ATP11A, being involved in LPS-induced TRL4 signaling, although in a way that would be actually opposite to that suggested by Pineau and coll. results. Indeed, silencing of ATP11A was found to enhance proinflammatory cascade (NFkB, cytokine release).

Reviewer 3 Report

The authors have identified a DNA methylation site that acts as a biomarker for severe CF patients versus patients with mild to moderate form of CF. This is a preliminary finding and has been well executed given the constraints of a clinical study. This finding could be of clinical importance for screening existing CF patients - especially given the easy methodology for DNA methylation screening.

Specific comments:

  1. The intent of this study was to find a biomarker that is associated with a severe form of CF disease. Given that intent, the authors need to include a clearer discussion of the correlation of this new biomarker with the specific form of CF mutation and the treatment strategy that the patient is undergoing. This could be of tremendous benefit to the community of physicians. Ie. The results listed in Table 2 need to be discussed in detail and suggestions for follow up studies should be made.
  2. ATP11A is an ATPase that might play an important role in cell biology of CF disease. There are a number of large single cell sequencing studies available for the human lung – including onces in CF lung. The authors should map the expression of ATP11A and talk about which cells it might be expressed in.
